# Therapeutic Potential of IL-1 Antagonism in Hidradenitis Suppurativa

**DOI:** 10.3390/biom14020175

**Published:** 2024-02-01

**Authors:** Laura Calabrese, Dalma Malvaso, Giulia Coscarella, Flaminia Antonelli, Alessandra D’Amore, Niccolò Gori, Pietro Rubegni, Ketty Peris, Andrea Chiricozzi

**Affiliations:** 1Dermatologia, Dipartimento di Medicina e Chirurgia Traslazionale, Università Cattolica del Sacro Cuore, 00168 Rome, Italy; laura.calabrese@unisi.it (L.C.); malvasodalma@gmail.com (D.M.); giuliacoscarella@gmail.com (G.C.); flamiantonelli@gmail.com (F.A.); niccologori89@gmail.com (N.G.); ketty.peris@unicatt.it (K.P.); 2Dermatology Unit, Department of Medical, Surgical and Neurological Sciences, University of Siena, 53100 Siena, Italy; pietro.rubegni@unisi.it; 3UOC di Dermatologia, Dipartimento di Scienze Mediche e Chirurgiche, Fondazione Policlinico Universitario A. Gemelli—IRCCS, 00168 Rome, Italy; aledamore@yahoo.it

**Keywords:** hidradenitis suppurativa, IL-1, IL-36, biologics, clinical trial, anakinra, bermekimab, spesolimab, imsidolimab, canakinumab

## Abstract

The immunopathogenesis of HS is partially understood and exhibits features of an autoinflammatory disease; it is associated with the potential involvement of B cells and the contribution of Th1 or Th17 cell subsets. Recently, the pathogenic role of both innate immunity and IL-1 family cytokines in HS has been deeply investigated. Several agents targeting the IL-1 family pathway at different levels are currently available and under investigation for the treatment of HS. HS is still characterized by unmet clinical needs and represents an expanding field in the current scientific research. The aim of this narrative review is to describe the pathological dysregulation of IL-1 family members in HS and to provide an update on therapeutic strategies targeting IL-1 family cytokine signaling. Further clinical and preclinical data may likely lead to the enrichment of the therapeutic armamentarium of HS with IL-1 family cytokine antagonists.

## 1. Introduction

Hidradenitis suppurativa is an inflammatory skin disease with a chronic course and a complex multifactorial pathogenesis. The disease affects approximately 1% of the general population [1]. Because of its peculiar clinical manifestations, consisting of nodules, abscesses, pus-discharging fistulae, and scars, mainly localized at skin folds, HS can detrimentally affect the quality of life of patients, influencing social, personal, and emotional life [2].

The immunopathogenesis of HS is only partially elucidated and seems to clearly differ from prototypic autoimmune dermatoses, simultaneously showing characteristics of an auto-inflammatory disease (inflammasome-driven dominance of interleukin-1β (IL-1β)), the presence of a possible involvement of B cells, and a contribution of Th1 or Th17 cell subsets [3]. 

Recently, the role of dysregulation of innate immunity and thus the IL-1 family pathway in HS has been explored. Indeed, there is strong evidence for the occurrence of HS in complex autoinflammatory syndromes caused by mutations in pattern recognition receptor (PRR) signaling pathways, including PASH, which consists of multiple neutrophilic dermatoses: pyoderma gangrenosum, acne, and HS [4]. Furthermore, the reported clinical success of IL-1 blockades (targeting IL-1α or IL-1β) provides further convincing evidence about the role of the IL-1 family in the pathogenesis of HS [5,6]. 

Despite the high burden of the disease on patients, the therapeutic options for HS are currently limited. Adalimumab (anti-TNF-α monoclonal antibody(mAb)) is the first FDA-approved biologic agent for HS, providing a single-cytokine blockade and a therapeutic response in approximately 50% of HS patients [7]. Recently, secukinumab, a mAb selectively binding to IL-17A, was approved for the treatment of adults with active moderate-to-severe HS [8]. In both the SUNSHINE and SUNRISE studies, a greater proportion of patients receiving secukinumab achieved HiSCR50 compared to placebo (SUNSHINE 41.3% vs. 29.4%; SUNRISE 42.5% vs. 26.1%) [9].

The aim of this narrative review is to describe the pathological dysregulation of IL-1 family members in HS and to provide an update on therapeutic strategies targeting IL-1 family cytokine signaling.

## 2. The IL-1 Family

The IL-1 family consists of a large group of cytokines, partially sharing a molecular structure, a common receptor-binding mode, and a similar signaling pathway [10].

The main function of the IL-1 family is to respond to tissue damage caused by pathogen-associated molecular patterns (PAMPs) or damage-associated molecular patterns (DAMPs) [11]. The family encompasses 11 cytokines and 10 receptors, with a mode of action and signal transmission that have not been fully elucidated [12].

Within the IL-1 family of cytokines, seven members exert agonistic effects on their receptors (IL-1α, IL-1β, IL-18, IL-33, IL-36α, IL-36β, and IL-36γ), and there are four members with antagonistic effects, such as IL-1 receptor antagonist (IL-1Ra), IL-36 receptor antagonist (IL-36Ra), IL-37, and IL-38. All members, albeit to varying degrees, are key molecules in mediating various immunological responses, primarily orchestrating innate immunity and bridging the gap between innate and adaptive immunity. IL-1α and IL-1β are encoded as full-length precursors and then processed by enzymatic cleavage, which is essential for IL-1β, but not for IL-1α activation [10,12]. These molecules are encoded by two different genes and share part of their sequence. However, due to the common overall folding, they bind to the same receptor complex, which is composed of the primary receptor IL-1R1 and IL-1 receptor accessory protein (IL-1RAP, also named IL1R3), thus forming a signal-competent ternary complex [13]. 

Recently described members of the IL-36 subfamily include IL-36α, IL-36β, IL-36γ, IL-36Ra, and IL-38, with the latter two playing an antagonistic role [14]. 

IL-36α, IL-36β, and IL-36γ are prominently expressed on the barrier sites of the body, particularly on the skin, and their aberrant signaling is involved in several inflammatory and infectious dermatoses [15]. 

One of the most recently described members within the IL-1 family was IL-33, which is considered a key cytokine of both innate and adaptive responses, particularly in allergic inflammation. As well as IL-1α, IL-33 can transmit its signal by acting as a transcription factor in the cell nucleus or, when secreted extracellularly, by binding its receptor IL-1R4 (ST2 or IL-33R) [16]. This results in the recruitment of IL-1R3 and in the generation of the ternary complex [12]. 

The IL-18 subfamily comprises IL-18, which was first described as an ‘IFNγ-inducing factor’, and IL-37. IL-18 binds to its receptor IL-1R5 (IL-18Rα), forming a binary complex that is recognized by the co-receptor IL-1R7 (IL-18Rβ) [17]. The resulting ternary complex is very similar to that involving IL-1β (IL-1β, IL-1R1 and IL-1R3). IL-37, which represents an anti-inflammatory protein, also binds to IL-1R5 [18], but the recruitment of IL-1R8 (TIR8 or SIGIRR) as a co-receptor enables the initiation of signal transduction [19].

## 3. Hidradenitis Suppurativa Pathogenesis

The pathogenesis of HS is partially elucidated, although it is known that a complex interplay between genetic, hormonal, immunological, and microbial factors, along with tobacco smoking and obesity, contribute to disease occurrence and/or severity [20].

The current model of HS pathogenesis includes an initial involvement of the pilosebaceous unit (PSU), leading to surrounding inflammation through a multistep process encompassing follicular occlusion and subsequent rupture [21]. However, whether the infundibular alteration or the immune cell infiltration is the *primum movens* has not been finally defined [20]. 

In the skin folds, mechanical friction is likely responsible for triggering the inflammatory cascade, together with nicotine stimulation and bacterial colonization (Figure 1) [3]. This results in the release of Damage and Pathogen Associated Molecular Patterns (DAMPs and PAMPs, respectively) and consequent activation of innate immunity. In detail, immune resident cells, such as macrophages, are hyperresponsive to PAMP stimulation in HS because of an increased expression of toll-like receptor 2 (TLR2) [22], secrete pro-inflammatory cytokines including TNFα, IL-6, and IL-1β. The inactive precursor of IL-1β (pro-IL-1β) is converted into the active form IL-1β through a cleavage process mediated by a multiprotein complex called inflammasome [23].

The inflammasome components NLRP3 and caspase-1 were found to be highly expressed in HS skin and in keratinocytes, which represent an additional source of IL-1β production in HS lesions [24].

Noteworthily, according to this model of HS pathogenesis, keratinocytes actively contribute to the HS inflammatory milieu through the secretion of pro-inflammatory cytokines such as IL-1β and IL-36α, IL-36β, and IL-36γ (belonging to the IL-1 cytokine family) [25].

IL-36 cytokines are neutrophil chemo-attractants [26,27]. Neutrophil-derived proteases such as cathepsin G, elastase, and proteinase-3 have been reported to be IL-36 activator enzymes, increasing the biological activity of IL-36 proteolytic cleavage of IL-36 cytokines. Infiltrating neutrophils contribute to inflammatory cytokine production and pus formation in HS [28,29].

Furthermore, TNF and IL-1β can stimulate endothelial cells and strengthen the expression of chemokines, such as CXCL1, CXCL6, CXCL8, CXCL11, CCL20, and CCL22, favoring the recruitment of neutrophils, T-cells, and monocytes into the skin [30]. 

IL-1β is also able to induce the production of an extracellular matrix, degrading enzymes such as matrix metalloproteinases (MMPs). These enzymes can facilitate the rupture of dilated hair follicles and promote the final destruction of the skin with abscess and tunnel formation [31].

In chronic stages of lesion development, CD4-positive T-cells that produce both IL-17 and IFNγ, indicative of a Th1/17 phenotype, are enriched in lesions together with high levels of the IL-1 family cytokines IL-1β and IL36 as well as TNFα [3,32].

In recent years, the dysfunction of the innate immune compartment in HS has increasingly gained attention, and emerging evidence shows a driving role of the IL-1 family in HS pathophysiology.

## 4. Evidence of the Role of IL-1 in Hidradenitis Suppurativa

### 4.1. Preclinical Evidence 

The first evidence about the role of IL-1 cytokines in HS was published in 2011 through the definition of the cytokine profile of lesional and perilesional HS skin cultured in a trans-well culture system [33]. Levels of IL-1β in a medium from 24 h cultured HS skin were found to be significantly elevated in comparison to those from both perilesional and healthy controls [34]. A similar experiment based on a trans-well culture system revealed high levels of IL-1β and IL-33 in culture media from HS skin compared with healthy controls [35].

Along these lines, increased mRNA expression of IL-17A, IL-1β, IL-10, and TNFα, measured through real-time polymerase chain reaction (PCR), were found in HS skin samples in comparison with perilesional, uninvolved, and healthy control skin. Furthermore, flow cytometry showed that the main source of IL-1β was a subset of CD11c+ CD1a- CD14+ cells [36]. The same study showed that both the mRNA and protein expression of IL-18 was significantly enhanced in HS lesional skin [36].

Results from another recent experimental study extensively supported the role of IL-1 in HS. In detail, real-time PCR detected 8-fold and 130-fold higher IL-1β expression in HS lesions than in psoriasis lesions and healthy control skin, respectively. In vitro IL-1β-stimulated cells (immune cells, microvascular dermal endothelial cells, dermal fibroblasts, and keratinocytes), analyzed through RNA-seq, showed an upregulation of IL-1β transcripts, mostly derived from fibroblasts [31]. 

Interestingly, the same study demonstrated that the strong IL-1β signature could be reversed by the administration of IL-1 receptor antagonist (IL-1Ra) [31]. 

Unlike IL-1, evidence on the contribution of IL-36 cytokines to HS pathogenesis is limited and their expression seems to be lower in HS than in psoriasis [37].

Immunohistochemical (IHC) analysis on HS skin samples showed a trend toward an elevated protein expression of IL-36α and IL-36RA as well as a significant upregulation of IL-36β and IL-36γ, even though levels were not as high as in lesional psoriasis skin [38]. Semiquantitative RT-PCR on HS skin samples demonstrated a statistically significant increase of IL-36α in comparison with healthy skin. Both IL-36β and IL-36γ showed a trend to elevation, while IL-36RA levels were comparable to healthy skin. Conversely, all IL-36 cytokines were significantly elevated in lesional psoriasis skin compared with healthy controls [38].

A similar study investigated the contribution of the IL-36 sub-family in HS lesional and perilesional skin in comparison with healthy control skin [39]. Expression levels of IL-36α, IL-36β, and IL-36γ, measured by RT-PCR and IHC staining, were all significantly higher in lesional HS skin than in healthy controls, with IL-36γ being the only cytokine significantly upregulated in lesional vs. perilesional HS skin. Interestingly, both IL-37 and IL-38 cytokines were significantly overexpressed in perilesional HS skin compared with healthy controls, though their expression in HS lesional was lower than in perilesional HS skin [39]. 

Furthermore, a gene expression study through RT-PCR showed that IL-36α, IL-36β, and IL-36γ were all enhanced in HS lesional skin compared with healthy control skin, with IL-36γ being the most upregulated cytokine. Conversely, IL36Ra was not found to be increased in HS skin [40]. In another study, serum levels of IL-36α, IL-36β, and IL-36γ, measured through enzyme-linked immunosorbent assay (ELISA), were found to be significantly higher in HS patients compared with healthy controls [41].

Concerning other members of the IL-1 family, one study detected decreased mRNA levels of IL-37 (with anti-inflammatory effects) in HS samples [42].

Overall, these experimental studies suggest the relevance of the IL-1 family in HS.

### 4.2. HS as an Autoinflammatory Disease

Autoinflammatory disorders (AIDs) encompass a heterogeneous group of entities featured by genetic defects and/or a documented dysregulation of key innate immune pathways, including an excessive IL-1 signaling [43].

Indeed, the occurrence of HS in the context of rare autoinflammatory syndromes, such as PASH (pyoderma gangrenosum, acne, and HS), which is considered to date as the paradigm of HS-related autoinflammatory syndromes within the spectrum of neutrophilic dermatoses, further support a key role of innate immune dysregulation in HS [44]. Other polygenic autoinflammatory syndromes that include HS, with a documented role for IL-1, are PAPASH (pyoderma gangrenosum, acne, pyogenic sterile arthritis, and HS) and PASS (pyoderma gangrenosum, acne vulgaris, ankylosing spondylitis, and HS) [44]. 

Moreover, mounting evidence suggests that HS could be even included in a newly recognized subtype of autoinflammatory diseases, named autoinflammatory keratinization diseases (AiKDs). AiKDs are characterized by the following three criteria: (i) the primary sites of inflammation are the epidermis and the upper dermis; (ii) inflammation leads to hyperkeratosis, which is the main and characteristic phenotype of AiKDs; and (iii) AiKDs have primary genetic causative factors associated with the hyperactivation of innate immunity (autoinflammation), mainly in the epidermis and upper dermis [45]. 

Indeed, several genes linked to the keratinization process (e.g., NCSTN, PSENEN, and GJB2), but also to autoinflammation (e.g., PSTPIP1, MEFV, MVK, NOD2, NLRP3, and IL1RN), have been reported in patients with HS and its syndromic forms, suggesting that both pathways are crucial in the pathogenesis of the disease [46]. 

Interestingly, the gene PSTPIP1 encodes an enzyme named proline–serine–threonine phosphatase-interacting protein 1 (PSTPIP1). Gain-of-function mutations of this gene result in hyperphosphorylation of PSTPIP1 and enhanced assembly of the pyrin inflammasome, with subsequent uncontrolled IL-1β release [47]. The term PSTPIP1 spectrum was recently coined to encompass a range of clinical conditions with shared genetic backgrounds and similar clinical manifestations, including HS and its syndromic forms PASH, PAPASH, and PASS [48].

Furthermore, recent reports highlight the association of HS with well-known monogenic and IL-1-driven autoinflammatory diseases, such as familial Mediterranean fever (FMF) [49,50,51] or mevalonate kinase deficiency (MKD) [52,53]. The coexistence of HS with monogenic AIDs entities with shared patho-mechanisms may therefore shed light on new and yet unexplored molecular pathways underlying autoinflammation.

### 4.3. Clinical Evidence

As a result of the increasing understanding of the central role of the IL-1 family pathway in HS pathogenesis, numerous targeted therapies are in the pipeline to explore the potential impact of IL-1 signaling blockade on the HS treatment spectrum (Table 1).

#### 4.3.1. Anakinra

Recently, selective blockade of the IL-1 pathway in HS with anakinra (recombinant IL-1R antagonist) has been investigated in an open-label pilot study (NCT01516749), demonstrating a clinical improvement in six patients with moderate-to-severe HS, as evidenced by a significant reduction in the modified Sartorius score and improvement in quality of life [54]. All participants were treated with 100 mg anakinra daily for 8 weeks, followed by 8 weeks off therapy. However, a recurrence of the disease was observed during the follow-up period. Injection site reactions, which were resolved in all patients by week 2 of treatment, were the only adverse events (AEs) reported. Overall, anakinra has a favorable safety profile, with increased risk of infection and neutropenia being the most commonly reported adverse events [55]. No increased risk of malignancy was reported. In addition, a double-blind, randomized, placebo-controlled clinical trial (RCT) conducted on 19 patients with moderate to severe HS (NCT01558375), showed efficacy of anakinra in 78% of patients [5]. No serious AEs were reported. 

André R et al. reported long-term efficacy and safety data of anakinra in three patients with HS (Hurley II and III) [56]. Two patients experienced extended efficacy over 3 and 7 years, respectively, with a favorable safety profile. 

#### 4.3.2. Bermekimab

Bermekimab, a human anti-IL-1α monoclonal antibody, was tested in an RCT (NCT04019041) in 20 patients with moderate-to-severe HS for whom anti-TNF therapy was ineffective or contraindicated. Furthermore, 60% of HS patients receiving 12 weeks of intravenous bermekimab achieved HiSCR50 compared to 10% in the placebo group, with clinical efficacy maintained in 40% of patients in the active arm through week 24, 12 weeks after drug interruption [57]. Improvements in patient-reported outcomes were observed in the bermekimab group.

A phase II clinical trial (NCT03512275) was subsequently designed to evaluate the safety, tolerability, and efficacy of 400 mg weekly for 13 consecutive weeks (week 0 to week 12) in patients with moderate-to-severe HS who had previously failed anti-TNF treatment (group A) and those who were naïve to anti-TNF therapy (group B) [6].

The primary endpoints of the study were safety and tolerability. The most common AEs were injection site reactions and nausea, both of which were mild to moderate in severity. Overall, the drug was well tolerated in all patients throughout the study. At week 12, 63% and 61% of patients in groups A and B, respectively, achieved HiSCR50. Significant improvements were similarly reported in other secondary outcomes, including a Physician’s Global Assessment (PGA) and patient-reported outcomes, highlighting the potential role of bermekimab in treating HS patients who are refractory to adalimumab or for whom the drug is contraindicated. 

A further trial was terminated as it met pre-established futility criteria (NCT04988308).

#### 4.3.3. MEDI8968 (AMG 108)

The drug, a fully human monoclonal antibody that selectively binds to IL-1 receptor 1 (IL-1R1) and inhibits its activation by both IL-1α and IL-1β, was evaluated in a phase II trial to provide preliminary evidence of the safety, tolerability, and efficacy of MEDI8968 in subjects with moderate to severe HS, as assessed by the proportion of subjects achieving PGA 0, 1, or 2 at week 12 (NCT01838499). A total of 224 patients were enrolled, but the response rate was only 23.6% in the active arm compared to 18.5% in the placebo arm. Consequently, the study was terminated prematurely due to lack of efficacy [58].

#### 4.3.4. Canakinumab

Currently, data concerning the use of canakinumab, a fully humanized monoclonal antibody targeting IL-1β, in the treatment of HS are scarce and derive mainly from case reports and series [59,60,61,62]. Results are conflicting, with patients experiencing partial or poor responses. Regarding the safety profile, some patients reported AEs including infections, gastrointestinal disorders, headache, and dizziness. Investigating the efficacy and safety of canakinumab in a larger cohort may provide a clearer understanding of its potential role in HS management. 

#### 4.3.5. Lutikizumab

Lutikizumab (ABT-981) is a dual variable domain immunoglobulin that potently neutralizes both IL-1α and IL-1β. Currently, a phase 2 study is evaluating lutikizumab in 160 patients unresponsive to anti-TNF therapy (NCT05139602). Participants will receive subcutaneous injections of lutikizumab (ABT-981) or placebo for 16 weeks to assess the percentage of subjects achieving HiSCR50 at this endpoint. The secondary outcome is the percentage of participants achieving an improvement in pain as measured by the Numeric Rating Scale (NRS) of 30 in participants with a baseline of NRS ≥ 3.

#### 4.3.6. Zimlovisertib and KT-474

Two compounds targeting interleukin-1-associated kinase 4 (IRAK4) are being investigated for their potential role in HS therapy. Zimlovisertib (PF-06650833) is a highly potent and selective small molecule inhibitor of IRAK4 that concluded a phase II multiple kinase inhibitor versus placebo trial (NCT04092452). Patients were randomized in three active arms or placebo for a 16-week dosing period and 4 weeks of follow-up to assess the percentage of subjects achieving HiSCR50 at week 16. Preliminary data on 400 mg of zimlovisertib once daily highlighted that 16/47 (34%) of patients achieved HiSCR50 compared to baseline [63]. AEs included headaches (11%), acne (9%), urinary tract infections (6%), HS worsening (2%), and nausea (2%).

KT-474 is an orally administered small molecule that has completed a phase I trial in 154 healthy volunteers, HS patients, and atopic dermatitis (AD) patients (NCT04772885), with single and multiple ascending doses to evaluate the safety, tolerability, pharmacokinetics, and pharmacodynamics of the drug. A total of twelve patients with moderate-to-severe HS and seven AD patients concluded the four-week treatment period. Clinical endpoints collected included AN count, Pain NRS, Pruritus NRS, and HS-PGA. At week 4, 42% of participants achieved HiSCR50 and 25% attained HiSCR75, with a mean reduction in AN count of −46% [64]. KT-474 has a favorable safety profile with no serious AEs emerging and no drug-related infections. AEs included headaches, fatigue, and diarrhea and were rated predominantly as mild. 

The promising early efficacy and safety data supported the development of a phase II clinical trial of KT-474 to further investigate its potential therapeutic role in HS (NCT06028230).

#### 4.3.7. MAS-825

A monoclonal antibody inhibiting IL-1α and IL-18 named MAS-825 is under clinical investigation. MAS-825 and three other compounds are being evaluated in a phase II trial to assess their efficacy and safety compared to placebo (NCT03827798). 

#### 4.3.8. Spesolimab

Selective inhibition of IL-36 cytokines (also members of the IL-1 family) is being explored with several molecules including spesolimab, a mAb targeting IL-36 receptor, currently under investigation in clinical trials for moderate-to-severe HS. In a phase II proof-of-concept study, patients (n = 52) were randomized 2:1 to receive spesolimab or placebo intravenously once a week for 3 weeks, followed by subcutaneously every 2 weeks for 12 weeks (NCT04762277) [65]. At week 12, a decrease in all HS lesions, including draining tunnels (dT), abscesses, and painful nodules, was assessed in the active arm. 66.7% of participants showed a reduction in dT compared to 38.5% in the placebo group. 

Spesolimab was generally well tolerated, which is consistent with previous studies in other indications. In the open-label extension (OLE) study (NCT04876391), efficacy with spesolimab continuous treatment was maintained through week 24, with sustained reductions in lesion count and a percentage change in dTs and IHS4 score [66]. 

Headaches, nasopharyngitis, nausea, fatigue, and injection site reactions were the most frequently reported AEs, with no serious AEs occurring, which is consistent with previous studies in other indications.

Based on these results, spesolimab is being further investigated in a phase II/III trial (NCT05819398).

#### 4.3.9. Imsidolimab

The clinical trial evaluating imsidolimab, a mAb that selectively antagonizes IL-36 receptor, was discontinued after the phase II trial failed to demonstrate efficacy over placebo in meeting the study endpoints (NCT04856930) [67]. A total of 149 patients with moderate to severe HS were enrolled and received monthly doses of imsidolimab or placebo for a 16-week treatment period followed by a 16-week extension period. The mean baseline AN lesion counts for the high-dose imsidolimab group, the low-dose imsidolimab group, and the placebo arm were 14, 11.9, and 12.1, respectively; the mean baseline dT counts for the high-dose imsidolimab group, the low-dose imsidolimab group, and the placebo group were 4.1, 2.7, and 3.1, respectively.

The primary endpoint was the mean change from baseline in AN lesion count at week 16. The high-dose imsidolimab group achieved a mean change of −5.9, compared to −4.1 in the low-dose imsidolimab and −5.6 in the placebo groups. At week 16, 41% and 39% of patients in the high- and low-dose arms, respectively, achieved HiSCR50 (vs. 35.7% placebo).

As concerns the safety profile, the most common AEs observed in imsidolimab and placebo-treated patients were COVID-19 (n = 10) and HS worsening (n = 8), both mild to moderate in severity

## 5. Discussion

Nowadays, IL-1 has emerged as a central mediator in autoinflammation, and the number as well as the complexity of IL-1-mediated autoinflammatory diseases has increased. 

Dysfunction of the IL-1 family cytokines is involved not only in monogenic AIDs but also in more common polygenic disorders, including HS. Indeed, experimental studies showed a strong contribution of innate immunity to HS pathophysiology [68]. Advances in research into the role of the IL-1 cytokine family have provided a rationale for considering IL-1 antagonism as a viable therapeutic strategy for the treatment of HS.

Several agents targeting the IL-1 family pathway at different levels are currently available and under investigation for the treatment of HS (Figure 2). 

Anakinra is a recombinant IL-1Ra that simultaneously inhibits IL-1α and IL-1β [69,70]; rilonacept [71] is a recombinant soluble decoy receptor of IL-1β that neutralizes IL-1β and, with lower affinity, IL-1α and IL-1Ra; canakinumab [72,73] is a monoclonal antibody targeting IL-1β; lutikizumab (ABT-981) is a dual variable domain immunoglobulin that inhibits both IL-1α and IL-1β.

Anakinra has shown relatively good results in HS, as demonstrated in a small open-label study and in a randomized clinical trial [5,54]. Conversely, data on the efficacy of canakinumab in HS, from case reports and series, showed contrasting results [59,61]. 

Additional mAbs’ targeting the IL-1 pathway have been investigated in clinical trials for HS: bermekimab (anti-IL-1α), gevokizumab (anti-IL-1β), MEDI8968 (AMG 108; anti-IL-1R1), MAS-825 (anti-IL-1α and IL-18), and RPH-104, a heterodimeric fusion protein that inhibits IL-1β. Among these, bermekimab has already shown clinical efficacy in patients with HS, both in a randomized clinical trial and in an open-label study [6,57]. An alternative mechanism of action is the inhibition of IRAK4 kinases, which are involved in modulating the innate immune response and are therefore a potential therapeutic target in immune-mediated diseases. Two compounds targeting IRAK4, zimlovisertib, and KT-474, are being investigated. Preliminary efficacy data from the phase II trial did not show promising results, with 34% of patients achieving HiSCR50 at week 16 [63]. 

IL-36 antagonism has recently proved to be successful in the treatment of another autoinflammatory disease, namely generalized pustular psoriasis [73]. Based on the overexpression of IL-36 cytokines detected in HS, two agents selectively targeting the IL-36 receptor, spesolimab and imsidolimab, have been selected for clinical investigation. The results of these trials will provide meaningful information on whether IL-36 antagonism may also be an encouraging therapeutic strategy in HS. To date, results from a phase II proof-of-concept study, showing promising efficacy and safety profiles, support the development of spesolimab in HS [66].

Overall, the drugs were found to be safe and well-tolerated. Most adverse events that occurred in clinical trials were mild to moderate in severity and included headache, nausea, fatigue, injection site reactions, infections, and worsening of HS.

No antagonists of other members of the IL-1 family have been tested in HS so far.

In conclusion, HS owns unmet clinical needs and represents a stimulating field for scientific research. Further clinical trials will hopefully lead to the addition of IL-1 family cytokine antagonists to the HS treatment armamentarium in the near future.

## Figures and Tables

**Figure 1 biomolecules-14-00175-f001:**
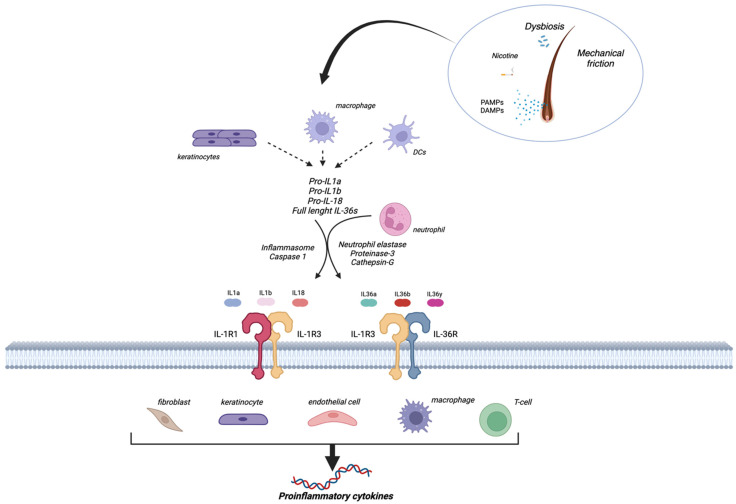
The pathogenetic role of IL-1 family cytokines in hidradenitis suppurativa. The inflammatory cascade in HS is triggered by various external stimuli such as smoking, dysbiosis, or mechanical stress. This results in the release of DAMPs and PAMPs and the consequent activation of innate immunity. Activation of macrophages, keratinocytes, and dendritic cells, leads to the production of pro-inflammatory cytokines, including interleukin-1 (IL-1), IL-18, IL-36, TNFα, and IL-6. The inactive precursor of IL-1β (pro-IL-1β) is converted into the active form IL-1β through a cleavage process mediated by the inflammasome. Neutrophil-derived proteases such as cathepsin G, elastase, and proteinase-3 have been reported to be IL-36 activator enzymes. After proteolytic cleavage, the precursors are converted to active forms and bind to their receptor complexes, initiating an inflammatory cascade that ultimately leads to the destruction of the pilosebaceous unit. DAMPs, damage-associated molecular pattern molecules; DC, dendritic cell; IL, interleukin; IL-R, interleukin receptor; PAMPs, pathogen-associated molecular pattern molecules.

**Figure 2 biomolecules-14-00175-f002:**
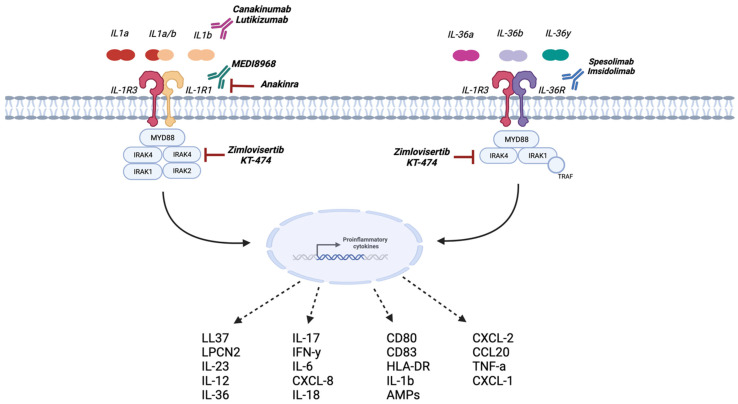
Therapeutic potential of anti-IL-1 agents in hidradenitis suppurativa. Several targeted agents are currently available or under investigation for the treatment of HS, investigating the potential benefits of blocking the IL-1 signaling pathway at various levels. IL-1 and IL-36, by binding their respective receptor complexes, are able to induce the expression of downstream genes, mostly transducing proinflammatory mediators. Two monoclonal antibodies canakinumab and lutikizumab primarily target IL-1β. MEDI8968 and anakinra exert their action by inhibiting the IL-1R1 receptor, thus preventing its activation by both IL-1α and IL-1β. Spesolimab and imsidolimab are two agents selectively targeting the IL-36 receptor. Lastly, zimlovisertib and KT-474 target IRAK4, thereby blocking the downstream inflammatory cascade. AMPs, antimicrobial peptides; CCL, chemokine (C-C motif) ligand; CD, cluster of differentiation; CXCL, chemokine (C-X-C motif) ligand; HLA-DR, human leucocyte antigen-DR isotype; IL, interleukin; IFN, interferon; IRAK, interleukin receptor-associated kinase; MYD, myeloid differentiation primary response 88; TNF, tumor necrosis factor; TRAF, tumor necrosis factor receptor-associated factor.

**Table 1 biomolecules-14-00175-t001:** Emerging IL-1 family targeting agents for hidradenitis suppurativa.

Drug Name	MoA	Study	Clinical Trial No.	Status	Primary Endpoint(s)
Anakinra	Recombinant IL-1R antagonist	Phase II, randomized, double blind, placebo controlled	NCT01558375	Completed	Efficacy at week 24
		Phase II, open label, non- randomized, proof-of-concept trial	NCT01516749	Completed	Changes in modified Sartorius score at week 8
Bermekimab (MABp1)	Anti IL-1α mAb	Phase II, randomized, double blind, placebo controlled	NCT04019041	Completed	Percentage of participants with HiSCR50 at week 12
		Phase II, randomized, double blind, placebo and active comparator controlled, dose ranging	NCT04988308	Terminated	Part 1. Percentage of participants with HiSCR50 at week 16 Part 2. Percentage of participants with HiSCR50 at week 12
		Phase II, open label	NCT03512275	Completed	Number of adverse events up to visit 14 (day 93)
MEDI8968 (AMG 108)	IL-1R1 inhibitor mAb	Phase II, randomized, double blind, placebo controlled	NCT01838499	Terminated	Percentage of subjects achieving PGA score 0, 1 or 2 from baseline to week 12
Lutikizumab	Anti IL-1 α/β mAb	Phase II, randomized, double blind, placebo controlled	NCT05139602	Active, not recruiting	Percentage of participants with HiSCR50 at week 16
Zimlovisertib (PF-06650833)	IRAK4 inhibitor, small molecule	Phase II, randomized, double blind, placebo controlled	NCT04092452	Completed	Percentage of participants with HiSCR50 at week 16
KT-474 (SAR444656)	IRAK4 inhibitor, small molecule	Phase I, randomized, placebo controlled, single and multiple ascending dose	NCT04772885	Completed	Incidence and severity of TAEs. Incidence and frequency of use of concomitant medication
		Phase II, randomized, double blind, placebo controlled	NCT06028230	Not yet recruiting	Percent change from baseline in AN count at week 16
MAS-825	Anti IL-1 β/IL-18 mAb	Phase II, randomized, double blind, placebo controlled	NCT03827798	Recruiting	Percentage of participants with HiSCR50 at week 16
Imsidolimab	IL-36R antagonist	Phase II, randomized, double blind, placebo controlled	NCT04856930	Completed	Percent change from baseline in AN count at week 16
Spesolimab	IL-36R antagonist	Phase II, randomized, double blind, placebo controlled	NCT04762277	Completed	Percent change from baseline in AN count at week 12
		Phase II, open-label, long-term extension study	NCT04876391	Active, not recruiting	Occurrence of TEAEs up to the end of maintenance treatment period including residual effect period (REP)
		Phase II/III, randomized, double blind, placebo controlled	NCT05819398	Recruiting	Percent change from baseline in draining fistulas/tunnels (dT) at week 8

AN, total abscess and inflammatory nodule count; HiSCR, Hidradenitis Suppurativa Clinical Response; IL, interleukin; IL-1Ra, interleukin-1 receptor antagonist; MoA, mechanism of action; PGA, Physician Global Assessment; TEAEs, Treatment-Emergent Adverse Events.

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
