# Peer review of "Therapeutic Potential of IL-1 Antagonism in Hidradenitis Suppurativa"

_biomolecules, 2024, doi:10.3390/biom14020175_

Round 1
Reviewer 1 Report
Comments and Suggestions for Authors
The article titled "Therapeutic potential of IL-1 antagonism in hidradenitis suppurativa is an interesting and well-written piece that explores the potential benefits of IL-1 antagonism in the treatment of HS . The authors provides a comprehensive overview of the current understanding of HS and the role of IL-1 in its pathogenesis. The article also discusses the promising results of recent studies that have investigated the use of IL-1 antagonists as a therapeutic approach for HS. Overall, this article is informative, engaging, and should be published without any revisions.
Author Response
Kind reviewer, thank you very much for your valuable comment.Reviewer 2 Report
Comments and Suggestions for Authors
The authors of the article point out IL-1 antagonism serve as feasible therapeutic strategies for the treatment of HS through review of the role of IL-1 family in autoinflammation.
This study has certain clinical significance and provides a lot of information. The topic is relevant in the field.
The authors of the article proposed the therapeutic strategy of IL-1 antagonists in HS by providing a large number of reviews. No antagonists of other members of the IL-1 family have been tested in HS so far.
The conclusions consistent with the evidence and arguments presented
and they address the main question posed. References and others were not found to be inappropriate.
The authors should summarize and look forward to the side effects that occur during the use of IL-1 antagonists.
Less...
Author Response
Thank you for your very interesting comment. A dedicated paragraph has been added to the discussion section.
Reviewer 3 Report
Comments and Suggestions for Authors
This is a nice review covering the disease hidradenitis suppurativa with focus on the potential role of IL-1 in the pathogenesis and as target for treatment. I have a few minor comments.
1. On page 2, line 91 the authors write "INFγ-inducing factor". Please change to "IFNγ-inducing factor".
2. On page 4, line 162 please specify the specific member of IL-17 that is referred to. Not only IL-17. Is it IL-17A or IL-17B or....
3.On page 6, line 235 please correct autoinflammaory to autoinflammatory.
4. On page 7, line 268 please specify the specific HiSCR that is referred to. Is it HiSCR50 or 75? This is also the case on page 8, line 279.
5. On page 11, line 417-418 the authors write "IL-36 antagonism has recently....generalized pustular psoriasis". Please include a reference for this statement.
Comments on the Quality of English LanguageThere are a few type errors and grammatical errors in the manuscript that needs to be corrected.
Author Response
Thank you very much for your suggestions. The authors have carried out the suggested modifications.